# Psychological experience of patients with postpartum depression: A qualitative meta-synthesis

Wu Jiaming[1]◉, Guo Xin[1]◉, Du Jiajia[1]‡, Peng Junjie[1]‡, Hu Xue●[2]*, Li Yunchuan[1]‡, Wu Yuanfang[3]‡

1 School of Nursing, Yunnan University of Chinese Medicine, Kunming, Yunnan, China, 2 Pediatric, Yan'an Hospital Affiliated To Kunming Medical University, Kunming, Yunnan, China, 3 Gastroenterology, Yan'an Hospital Affiliated To Kunming Medical University, Kunming, Yunnan, China

◉ These authors contributed equally to this work.
‡ DJ, PJ, LY and WY also contributed equally to this work.
* 1554681053@qq.com

## Abstract

### Purpose

To determine the psychological experiences of patients with postpartum depression by systematically reviewing, retrieving, and synthesizing data from qualitative studies.

### Methods

Nine databases were systematically searched for relevant publications, from database establishment to September 20, 2024. All qualitative studies in English and Chinese that explored the real-life experiences, feelings, etc., of individuals affected by postpartum depression after childbirth were included. The qualitative meta-synthesis was performed following the Preferred Reporting Items for Systematic Reviews and Meta-Analyses (PRISMA) recommendations. Two independent reviewers selected these studies and evaluated the quality of each study, conducting a meta-analysis to integrate the results.

### Results

A total of 28 studies revealed 12 sub-themes and 3 descriptive themes: negative physical and psychological experiences and coping strategies, role transition discomfort and impact, lack of relevant support.

### Conclusion

The prominence of negative physical and psychological experiences, the discomfort of role transformation, and the lack of relevant support in patients with postpartum depression. In future studies, more attention should be paid to maternal mental health, and full mental health screening during pregnancy and childbirth should be implemented. Psychological

**Data Availability Statement:** All relevant data are within the manuscript and its Supporting information files.

**Funding:** This study is supported by the Science and Technology Innovation Fund from the School of Nursing, Yunnan University of Chinese Medicine (YZHCKY2406) .Yunnan University of Traditional Chinese Medicine School of Nursing provided financial support for this study.

**Competing interests:** The authors have declared that no competing interests exist.

counseling services should also be provided to pregnant women. [REGISTRATION: PROSPERO ID: CRD42024504506].

# 1 Introduction

Postpartum depression is one of the most common health problems among postpartum women worldwide, causing distress and serious negative impacts on mothers and their families, and has received widespread attention in various countries [1]. The clinical symptoms are usually characterized by depression, frustration, sadness, crying, irritability, and poor coping skills. In severe cases, psychotic disorders characterized by a range of symptoms such as hallucinations or suicidality may occur [2]. The reported incidence of postpartum depression varies widely both within and outside the country due to differences in cultural background, assessment tools, diagnostic methods, ethnicity, time of assessment, and scale thresholds taken in different countries [1–3]. Some studies have reported that the global average incidence of postpartum depression is 17.2% [4]. According to the World Health Organization, globally, about 10.0% of pregnant women and 3.0% of mothers experience depression each year, with postnatal depression being more severe in developing countries, where the prevalence reaches 15.6 to 19.8% [5].

The harm of postpartum depression is multifaceted, such as being unfavorable to the mother's physical recovery. Family members are prone to misunderstandings and barriers, and severe depression may even threaten the health of the mother and baby [6]. The early recognition of postpartum depression is extremely important because of the low expectations and awareness of maternal health management of postpartum depression, as well as the many problems with screening and management of postpartum depression [7]. Currently, there has been a gradual increase in the number of qualitative studies on the psychological experience of postpartum depression patients. However, the understanding, expression, and psychological experience of postpartum depression vary among mothers from different countries, races, and cultural backgrounds. In addition, there is a lack of consensus on the psychological experience of maternal postpartum depression across studies, and the results of a single qualitative study do not systematically reflect the full picture of their psychological experience. Therefore, this study aims to comprehensively collect qualitative studies related to the psychological experience of postpartum depression patients, further analyze and summarize the results of qualitative studies, and comprehensively interpret the psychological feelings and experiences of postpartum depression patients. This will provide a reference basis for clinical workers to formulate relevant nursing strategies.

# 2 Methods

## 2.1 Design

This study was conducted based on the updated system review and meta-analysis guidelines of the Preferred Reporting Project (PRISMA) [8]. It used meta-synthesis methods to combine and present qualitative research results [9]. According to Sandelowski et al. [10], qualitative meta-synthesis is the process of aggregating, analyzing, and comparing the results of multiple studies to obtain more comprehensive, objective, and accurate research conclusions based on an in-depth understanding of their ideas and deeper meanings through certain statistical and logical methods. According to the description by Thomas and Harden [11], the researcher systematically searches for relevant literature and repeatedly reads and understands the original

study, analyzes and interprets the meaning of the findings of the original literature, and further summarizes the findings based on their meanings. This process leads to the formation of new sub-themes, which are then summarized into new descriptive themes. This meta-synthesis protocol is registered with PROSPERO (ID: CRD42024504506).

## 2.2 Literature inclusion and exclusion criteria

### 2.2.1 Inclusion criteria.

1. *Participant (P)*. Women suffering from a depressed mood in the postnatal period, that is, women who scored ≥10 on the Edinburgh Postnatal Depression Scale (EPDS) assessment or were diagnosed with postpartum depression [12].

2. *Interest of phenomena (I)*. Real-life experiences, feelings, etc., of being affected by postpartum depression after childbirth.

3. *Context (Co)*. Obstetrics outpatient clinics, psychiatric outpatient clinics, maternity wards in hospitals, mothers' homes, or monthly childcare center facilities.

4. *Study design (S)*. Different types of qualitative research, including rooted theory, phenomenology, descriptive research, etc.

**2.2.2 Exclusion criteria.** Duplicate publications; Incomplete information or inaccessible full-text literature; Literature not in English or Chinese; Reviews and academic papers; The study did not clearly indicate the score of EPDS or clearly diagnose postpartum depression in pregnant women; Literature with a C-level quality evaluation.

## 2.3 Literature search strategies

Nine databases, including PubMed, Cochrane Library, Web of Science, Embase, Medline, CNKI, VIP, CBM, and WANFANG, were selected for a systematic search. The search time ranged from database establishment to September 20, 2024. The search utilized Medical Subject Headings (MeSH) and free terms, including: pregnant women, woman pregnant, mothers, parturients, depression postpartum, postnatal depression, depression postnatal, post partum depression, postpartum depression, post natal depression, emotions, experience, feeling, need, demand, attitude, qualitative research, descriptive analysis*, interview, content analysis*, thematic analysis*, grounded theory, phenomenology, and qualitative study. To ensure a comprehensive literature search, Boolean operators were used to combine search terms and manual searches. Take PubMed as an example of a search formula (S1 File).

## 2.4 Literature screening and data extraction

Two researchers (first and second authors) with evidence-based training screened the literature and extracted information according to the inclusion and exclusion criteria (S2 File). The literature was initially screened by reading the title and abstract and then re-screened by reading the full text. In case of disagreement, the decision was discussed with the third researcher. Data extraction included authors, regions, research methods, subjects, phenomena of interest, and main findings.

## 2.5 Criteria for evaluating the methodological quality of literature

Included literature was independently evaluated by two researchers using the JBI Center for Evidence-Based Health Care Quality Evaluation Criteria for Qualitative Research [13]. In case

of disagreement, the decision was discussed with a third researcher. A-level is to fully meet the quality standards and minimize the possibility of bias, while B-level partially meets the quality standards with a moderate likelihood of bias. C-level does not meet the quality standards at all, and there is a high possibility of bias occurring. This study only included literature with a quality level of A or B.

## 2.6 Literature analysis methodology

We used meta-aggregation to synthesize the results of the qualitative study [14]. The results of the literature were integrated using the pooled integration method recommended by the JBI Center for Evidence-Based Health Care [13]. Guided by qualitative research methods, researchers repeatedly read, deeply dissected, and interpreted the findings of the included literature and formed new sub-themes after combining similar findings. The sub-themes with certain connections were then synthesized into a new integrative theme, and the corresponding sub-themes were assigned to the integrative theme. When two researchers disagreed, the decision was discussed with the third researcher.

# 3 Results

## 3.1 Procedure for extraction

The preliminary search yielded 2,332 documents. After removing duplicates, 1,233 documents were obtained. Further reading of the titles, abstracts, and full texts of the documents resulted in obtaining 88 documents. These documents were selected by excluding those that did not conform to the content of the study and the subject matter, as well as those that could not be accessed in full text. Finally, after quality evaluation, 28 documents were included (Fig 1).

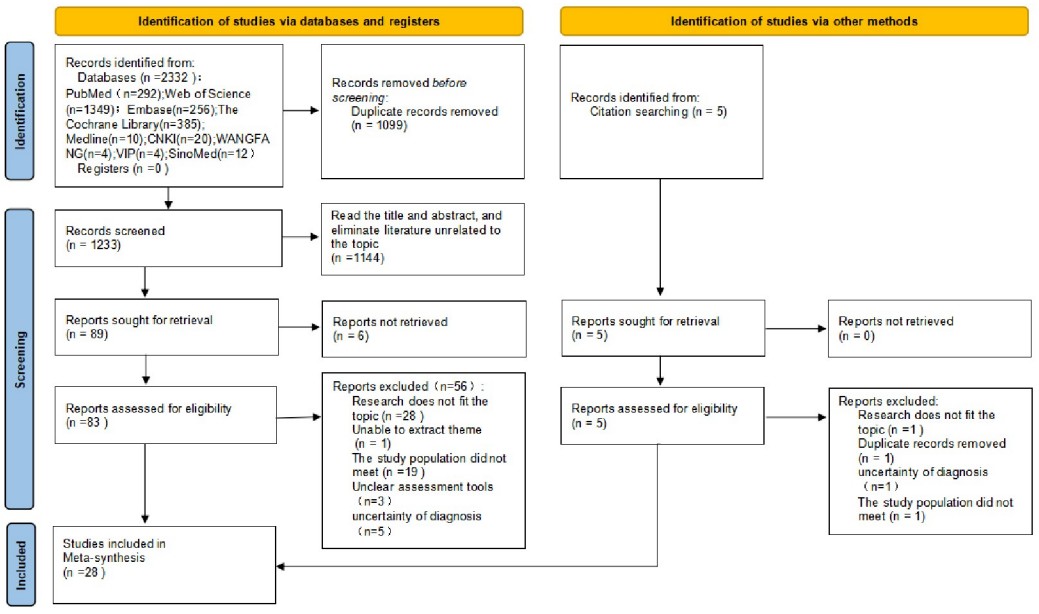

**Fig 1. PRISMA flowchart.**

## 3.2 Basic characteristics of the included literature and results of quality assessment

These 28 studies [15–42] had a total of 409 patients with postpartum depression. A variety of research methods were involved, including 13 phenomenological studies, 6 descriptive qualitative studies, 4 interpretive phenomenological studies, 3 grounded theory studies, and 2 studies that did not describe methods. The extraction results and the results of literature quality assessment are shown in Table 1. In this study, 12 new sub-themes were identified based on PRISMA and synthesized into 3 descriptive themes, as shown in Table 2.

## 3.3 Themes

**3.3.1 Meta-theme 1: Negative physical and psychological experiences and coping strategies.** *Sub-theme 1*: *Negative psychological experiences*. Postpartum depression is a common psychological problem. We found that most studies reported a variety of negative psychological experiences for participants [15, 19–21, 24–30, 32–38, 40–42]. On the one hand, after giving birth, women are prone to sadness, depression, and a series of cognitive changes, such as the inability to accomplish a task with sustained concentration and a lack of joy and confidence in life. Frustration arises after childbirth because of her own lack of ability and the characteristics of her personality, etc., and she feels lonely and helpless because of her family's lack of understanding, and she cannot control her emotions or thinking. In addition, mothers may experience conflicting feelings of guilt and responsibility for their children. On the other hand, the change in body image after delivery, the fear of not being able to return to normal life, and the beginning of closure of themselves. For example:

"I don't know why, I just want to cry." [27]

"I used to love. . . but now they don't make me happy, and I'm losing interest in these hobbies." [26]

"I can't control it; it's a terrible thing. I feel trapped. I feel like there's absolutely no way to get rid of this clock. No matter how hard I try, these terrible feelings won't disappear." [42]

"Yeah, that was another thing I was blaming myself for. I can't even feed my own baby." [21]

"But there were times when it was like the baby was crying. So what? I can't deal with this. Can't deal with it. . ." [40]

"Looking at a flabby waist and a bloated face, I prefer to isolate myself from others." [26]

*Sub-theme 2*: *Physical burden*. Mothers usually experience somatic stresses after childbirth [15, 19, 23–26, 29, 30, 32, 33, 35, 40, 41], such as postpartum contraction pains, incisional pains, headaches, nipple pains, difficulties in breastfeeding, difficulty in sleeping with reduced breast milk production, or insomnia. These stresses can aggravate postpartum depression, lead to somatic exhaustion, and create an uncomfortable somatic experience for the mothers. For example:

"I had a normal labor, but the baby's head was too big to be born, so I had a C-section. Now I can't turn over because of the pain in the incision." [17]

**Table 1. Key characteristics and quality appraisal of 28 included studies.**

| References, year | Country | Research method | Sample size | Interest of phenomena | Results | Quality category |
|---|---|---|---|---|---|---|
| Hanach et al. [15], 2024 | United Arab Emirates | Descriptive qualitative analyses | 27 | Explore the perceived mental health experiences and needs of mothers during the postpartum period | Six themes: Distinct postpartum experiences of primiparous and multiparous mothers; Experiences of emotional distress in the initial postpartum stage; Multifaceted challenges in breastfeeding; Multifactorial influences on postpartum mental health; Postpartum social support resources and providers; The need for formal and informal resources. | A |
| Borrero et al. [16], 2024 | USA | Descriptive qualitative analyses | 8 | Understand the perceptions of new mothers about factors contributing to their healthcare decision-making, for themselves and for their children, while living with postpartum depression | Three themes: Importance of clinician trust and support; Balancing the health of the mother and child; Other support structures that facilitate healthcare decision-making for the mother and baby dyad. | B |
| Liu et al. [17], 2022 | China | Phenomenological approach | 16 | Exploring the influences of postpartum depression in elderly mothers Factors | Four themes: Somatic stressors; Negative psychological experiences; Lack of social support and emergence of conflicts; The impact of role transformation on advanced maternal age | B |
| Maria et al. [18], 2021 | Australia | Phenomenological approach | 20 | Exploring the impact on physical activity and screen time in postpartum women with increased depressive symptoms | Seven themes: Individual level physical activity influences; Social level physical activity influences; Physical environmental level physical activity influences; Individual level screen time influences; Social level screen time influences; Strategies to increase physical activity; Strategies to reduce screen time | B |
| Atuhaire et al. [19], 2021 | Uganda | Phenomenological approach | 30 | Exploring the lived experiences of women recovering from a diagnosis of postpartum depression in southwestern Uganda | Five themes: somatic experiences; difficulties in home and family life; negative emotions; feelings of suicide, homicide and self-harm; coping with postpartum depression | B |
| Khalid et al. [20], 2020 | UK | Interpretative phenomenological analysis | 6 | To gain an in-depth understanding of the lived experience of women with postnatal depression | One theme: the conflicted mother | B |
| Tyson et al. [21], 2020 | Ireland | Descriptive qualitative analyses | 6 | Exploring the experience of a mom with postpartum depression | Four themes: Starting on a bad note; Support falling away; No space for mother's mind; Unwelcome familiar feelings | B |
| Liu et al. [22], 2019 | China | Interpretative phenomenological analysis | 14 | Understanding and exploring the psychological experience of women with postpartum depression | Three themes: Marital relationship; Mother-in-law/daughter-in-law relationship; Lack of childcare experience | B |
| Wang et al. [23], 2019 | China | Phenomenological approach | 12 | Understanding the psychological needs of patients with postpartum depression | Three themes: Support needs of consistent culture among family members; Obtainment needs for care knowledge; Anticipation needs for medical and nursing workers in obstetrics | B |
| Zhang et al. [24], 2019 | China | Phenomenological approach | 12 | Understanding the Real Experience of People with Postpartum Depression | Four themes: The true feelings as a new mother; Lack of spiritual and material support; Depression was more relevant with the type of melancholy character and peace character; The pressure from traditional culture of China. | B |
| Kazemi et al. [25], 2018 | Iran | Not stated | 16 | To understand the complex phenomenon of postpartum depression, learn more about the dimensions of the phenomenon, and explore the experiences of mothers with postpartum depression. | Three themes: Experienced problems; Unmet expectations; Psychological distresses | B |

*(Continued)*

**Table 1.** (Continued)

| References, year | Country | Research method | Sample size | Interest of phenomena | Results | Quality category |
|---|---|---|---|---|---|---|
| Zhao [26], 2018 | China | Phenomenological approach | 11 | Understanding how people with postpartum depression really feel about their illness and health wellness | Three themes: Poor concentration, lack of joy and confidence in life; Significant mood changes; Multiple behavioral challenges | B |
| Tian et al. [27], 2018 | China | Phenomenological approach | 14 | To explore the psychological feelings and inner demand of puerperas with earlypostpartum depression in Chengde City | Four themes: Experiencing complex psychological emotions; Difficulty adapting to role changes; Intense life frustration; Lack of relevant support | B |
| Li et al. [28], 2015 | China | Phenomenological approach | 12 | To deeply understand the real experience of postpartum depression among the mothers of hospitalized prema-ture infants in NICU and their psychological needs. | Three themes: Experiencing complex emotions; Carrying psychological stress; Lack of psychological support | B |
| Edhborg et al. [29], 2015 | Sweden | Phenomenological approach | 21 | Explore and describe the experiences and concerns during the first 3-9months following childbirth of those mothers who showed depressive symptoms 2–3 months postpartum, in a rural area in Bangladesh | Three themes: Family dynamics; Living at the limits of survival; Role of the cultural context after childbirth | B |
| Gardner et al. [30], 2014 | China | Interpretative phenomenological analysis | 6 | To explore the lived experience of postnatal depression in West African mothers living in the UK | Five themes: Conceptualising postnatal depression; Isolation; Loss of identity; Issues of trust; Relationships as a protective factor | B |
| Williams [31], 2013 | USA | Not stated | 9 | To describe the experience of women who perceived themselves as having recovered from postpartum depression | Four themes: Prelude to recovery; Igniting recovery; Recovery recounted as a victory; Realizing recovery was achieved | B |
| Wittkowski et al. [32], 2011 | UK | Grounded theory | 11 | Exploring the experiences of postnatal depression among South Asian mothers living in the UK | Three themes: Internalising misery; Others will judge me and I feel on my own; I talk to my health professional and they don't understand | A |
| Gao et al. [33], 2010 | China | Phenomenological approach | 15 | Describe the experience of postpartum depression among first-time mothers in mainland China. | Three themes: Feeling drained; Perceiving oneself to be a Failure; Dissonance | A |
| Abrams et al. [34], 2009 | USA | Grounded theory | 19 | Exploring the Experiences and Understandings of Postpartum Depression in Low-Income Mothers | Five themes: Ambivalence; Caregiving overload; Juggling; Mothering alone; Real-life worry | A |
| Barr [35], 2008 | Australia | Interpretative phenomenological analysis | 11 | Explore what is it like to become a mother, and examine how postpartum depression impacts on maternal adaptation | Three themes: "Stuck" in the liminal phase; Symptomatology; Mechanical infant caring | B |
| Buultjens et al. [36], 2007 | Australia | Descriptive qualitative analyses | 10 | To capture the missing voices of mothers who are suffering postnatal depression | Three themes: Becoming a mother: what to expect?; The birth of the baby: the experience of hospital stay; Perceptions of causes and experiences of postnatal depression; Women's perceived social support | B |
| Leung et al. [37], 2005 | China | Phenomenological approach | 11 | Report a study of the lived experience of postpartum stress among depressed Hong Kong Chinese mothers | Five themes: Parenting competence; The expectation experience gap; Baby-minder arrangements; Childcare demands; Conflict with culture and tradition | A |
| Edhborg et al. [38], 2005 | Sweden | Grounded theory | 22 | Explore and describe how Swedish women with signs of postpartum depression two months postpartum experience the first months with their child | Three themes: Struggling with life related to the self; Struggling with life related to the child; Struggling with life related to the partner | B |
| Templeton et al. [39], 2003 | UK | Descriptive qualitative analyses | 17 | To describe the experiences of women suffering from postnatal depression in black and minority ethnic communities in Wiltshire, UK | Four themes: Issues Specific to Pregnancy and Birth; Issues Specific to Health Care; Issues Specific to Culture; Other Issues | B |

(*Continued*)

**Table 1.** (Continued)

| References, year | Country | Research method | Sample size | Interest of phenomena | Results | Quality category |
|---|---|---|---|---|---|---|
| Amankwaa et al. [40], 2003 | USA | Descriptive qualitative analyses | 12 | Describe the nature of postpartum depression among African American women | Five themes: Stressing out; Feeling down; Losing it; Seeking help; Feeling better | A |
| Chan et al. [41], 2002 | China | Phenomenological approach | 35 | To examine the lived experiences of a group of Hong Kong Chinese women diagnosed with postnatal depression | Four themes: Trapped in the situation; Ambivalent towards the baby; Uncaring husband; Controlling and powerful in-laws | B |
| Beck et al. [42], 1992 | USA | Phenomenological approach | 7 | Describe the essential structure of lived experience of postpartum depression | Eleven themes: Loneliness; Hope; Obsessive thoughts; Haunted; Loss of interest and purpose; Fear and guilt; Unable to concentrate; Negativity; Insanity; Loss of control of emotions; Insecurities | B |

"After delivering, my breasts started swelling. I tried using herbal medicine, but all in vain. I decided to go to. . . Right now, I use only one breast to feed my child and also buy milk because milk from one breast is always not enough. . ." [19]

"I felt tired all the time, I still do. Exhausted, actually. I can't pull myself out of it. I am so tired most of the time and have great difficulty staying awake. I even bought a book, what's it called, on babies, but I don't seem to be able to read even a paragraph on some days." [35]

*Sub-theme 3*: *Self-recovery from postpartum depression*. Several studies have reported how mothers achieve self-recovery from postpartum depression [15, 18, 19, 30–32, 34, 40–42]. On the one hand, they seek help from family members, peers, social networks, professionals, and also through improving their cognition, self-motivation, distraction, and satisfying spiritual needs. In addition, some women will insist on postnatal rehabilitation exercises and reduce screen time to alleviate postnatal depression symptoms. On the other hand, some women will realize the recovery of postnatal depression by ending their unfortunate marriages. For example:

**Table 2. Descriptive themes, sub-themes, and frequencies.**

| descriptive themes | frequencies | sub-themes | frequencies |
|---|---|---|---|
| Negative physical and psychological experiences and coping strategies [15, 18–21, 23–38, 40–42] | 24 | Negative Psychological Experiences [15, 19–21, 24–30, 32–38, 40–42] | 21 |
| | | physical burden [15, 19, 23–26, 29, 30, 32, 33, 35, 40, 41] | 13 |
| | | Self-recovery from postpartum depression [15, 18, 19, 30–32, 34, 40–42] | 10 |
| Role transition discomfort and impact [15–27, 29, 30, 33–42] | 25 | Role transition [15, 20, 24, 25, 27, 30, 33–36, 38, 41] | 12 |
| | | Life-style modification [15, 17–21, 24, 26, 27, 29, 30, 36, 38, 41, 42] | 15 |
| | | Family life events [15, 19, 22–27, 29, 33, 37, 38, 40, 42] | 14 |
| | | Impact of parenting [16] | 1 |
| | | Socio-cultural influences [22, 24, 29, 33, 37, 39, 40] | 9 |
| Lack of relevant support [15–30, 32–38, 40–42] | 26 | Lack of emotional support [15, 16, 19, 21–25, 27–30, 32, 34, 36–38, 40, 42] | 19 |
| | | Lack of practical support [15, 16, 27, 41] | 4 |
| | | Lack of information support [15, 21–23, 27–30, 40] | 9 |
| | | Lack of other support [15, 19, 25–30, 32, 34, 36, 40] | 12 |

"I never got any help, really. I would just call people that I knew. I had received counseling, marriage counseling prior to the birth of my second baby. . . And I would call my—my— she was a psychiatrist. I would call her occasionally during the course of that first year." [40]

". . . Just like now, this group discussion, we are all strangers, but it feels very therapeutic." [15]

"But then you have to recover; you have no choice after a while. I think once I started feeling like my old self, you have to say, "Ah, I don't want to feel like this anymore." And really just get your head back in the game. Um, I think it is up here (pointing to forehead) as far as just trying to change yourself. The way you look at it, not be ashamed of it, not, you know." [31]

"I think that social connections are really important because that's when screen time becomes a factor, especially when you're feeling isolated or alone. So, if you're able to encourage more group activities or social interactions, such as partnering up with someone, that would be a really good way to reduce screen time." [18]

"At first, I had thoughts of killing myself. Then, I decided that instead of killing myself, I would rather leave this marriage. I would rather look for work as a housegirl, earn some money that would take care of us, rather than keep suffering in his house to the extent of wanting to kill myself." [19]

**3.3.2 Meta-theme 2: Role transition discomfort and impact.** *Sub-theme 1*: *Role transition*. After childbirth, the role of the mother changes [15, 20, 24, 25, 27, 30, 33–36, 38, 41], and some mothers may initially find it difficult to adapt to their role as mothers, causing a conflict of roles after the birth of their babies. As time passes, some mothers seek help to adapt to the role of motherhood, and their mindset changes.

"Just gave birth to a child, I was very anxious. I have very serious hemorrhoids and pubic symphysis separation. The first night back from the hospital, the lower half of my body felt crippled, and I couldn't move. I experienced severe pain and had to call '120' to go to the hospital. . . I just gave birth to a child, and I'm not adapting well to the feeling of nightmares." [24]

"I feel like I'm still a kid myself and can't take care of my kids' food, drink, and sleep." [27]

"Well, we wanted her very much, and I knew it would be hard having a baby. I knew that it would take a little while to get into a routine and things would change, but I didn't think things would be as disrupted as they have been. I expected to be able to do most things I used to do pretty much straight away." [36]

*Sub-theme 2*: *Life-style modification*. With the birth of a child, the mother has to assume the role of a caregiver, and her original lifestyle is disrupted [15, 17–21, 24, 26, 27, 29, 30, 36, 38, 41, 42], affecting her normal work schedule and family life, limiting her social activities, forcing her to give up some of her hobbies, and making her life dull and uninteresting. In addition, financial expenses have increased significantly compared to the previous period, and the standard of living has declined. For example:

"Since having a baby, I feel like my life can be described as a military mess (bitter smile)." [27]

"Since then, it has been difficult to find another job. I solely depend on my husband, who has failed to provide us with basic needs like milk, clothes, food, and even paying the house rent. . . All this just stresses me out." [19]

"In the past, I always like to get together with my friends to chat and hang out, but now I prefer to be alone and don't want to hang out with other people, and over time, I even have some horrible thoughts, what's the point of living like this all day long, why don't I just die?" [26]

*Sub-theme 3*: *Family life events*. After the birth of a child, the focus of the family shifts to the child, and the mother receives considerably less attention, resulting in a reduced sense of self-presence [15, 19, 22–27, 29, 33, 37, 38, 40, 42]. The lack of effective communication between husband and wife makes it easy for misunderstandings to arise. In addition, as the lifestyle of the mother differs from that of her mother-in-law, and as there are differences in parenting styles or thinking between generations, failure to deal with the situation correctly will lead to strained family relations and aggravate the symptoms of postpartum depression. For example:

"My family used to revolve around me, but now the whole family revolves around the children, and everything is centered on the children (despondent look)." [27]

"When my husband gets angry, he uses physical violence against me and always verbally insults me." [25]

". . . when I have had difficult times here, I have been sitting and thinking that my husband lives a luxurious life, going to his office, while he, of course, is under enormous pressure at work. . ." [38]

"My mother-in-law is from Shandong, and I'm from Hunan. The diet is different, but she only cares about her son eating well. I sit on the moon every day, eating steamed buns and noodles. The dishes are cooked very salty, and I can't eat them!" [23]

*Sub-theme 4*: *Impact of parenting*. Some mothers, after learning that postpartum depression not only harms them but can also affect their child's health, will often neglect their own health needs in order to meet their child's health needs or will have to make difficult decisions for their child's sake [16].

"I just felt that I needed to suppress my needs to ensure that [my daughter] was taken care of. . ." [16]

"I'm very against medicine. . . um, I knew that I needed to do what was best for my older daughter. And I felt that my mood was affecting her mood." [16]

". . .We did it with our first, so that actually hasn't been hard. Um, it has been easier the second time around. The decision to take, um, I am on an antidepressant. The decision to take that was hard." [16]

*Sub-theme 5*: *Socio-cultural influences*. The cultural factors could be alleviating, deteriorating, or neutral in relation to postpartum depressive symptoms [22, 24, 29, 33, 37, 39, 40]. However, there are some bad traditional practices that may have a negative impact on the mother or the baby. Traditional beliefs that mothers should stay at home to take care of their children, and the fact that most families seem to favor boys, can put a great deal of pressure on mothers. In addition, in China, some of the traditional practices of "doing the month" are not conducive

to the physical and mental health of the mother.(The practice of "doing the month" is an important post-natal recovery custom in traditional Chinese culture, which carries deep care and unique wisdom for women's physical and mental health after childbirth. This custom has a long history and has been passed down for thousands of years and is still widely practiced in China and even in some overseas Chinese communities. During the menstruation period, new mothers adopt a series of specific lifestyles, dietary habits, and health care measures, aiming to promote the rapid recovery of the body, prevent postpartum illnesses, and at the same time, adjust the psychological state and adapt to the change of the role of motherhood).

> "...She questioned why I couldn't take care of the baby myself. She said that the baby was my own son, and I am the mother. Why didn't I care for him myself? She meant I didn't take up my own responsibility..." [37]

> "For the baby's gender, they look a little heavy, held in the heart. I am more or less in a bit of pain. I did not want a second child at that time. Their families very much wanted a boy, and now they are also very unhappy because of the birth of a daughter. If it were a boy, it would have been better, and the relationship between the mother-in-law and daughter-in-law would have been better!" [24]

> "The 'doing the month'... It is like being in prison for me to be confined at home. I was not allowed to do anything but lie in bed. It was so boring. You know, I was a career woman before the baby was born, but..." [33]

**3.3.3 Meta-theme 3: Lack of relevant support.**     *Sub-theme 1*: *Lack of emotional support*. Emotional support from family members (especially husbands), professionals, or peers is especially important when a woman is physically and mentally exhausted and emotionally volatile after giving birth [15, 16, 19, 21–25, 27–30, 32, 34, 36–38, 40, 42]. In this study, it was found that some mothers not only need to take care of their babies but also need to complete heavy housework after delivery, but their husbands do not care and even blame them. Some mothers were misunderstood when they sought professional help. Additionally, some of the mothers said that it would help them to vent their emotions if they could get support from their peers. For example:

> "Otherwise, I do most of the work like washing, cooking, sweeping, and mopping myself. And when I feel tired and sometimes fail to eat...Yes, that's when I think that maybe I should leave or sleep the whole day, but the baby's clothes need to be washed. Yet, my husband can never support me...not even to hold the baby like I do." [19]

> "I got answers from professionals like, 'There is nothing wrong with you, go back home and stop disturbing us. Basically, you are wasting our time.' And they were horrible. It was a doctor who said that to me. My husband was sitting with me that day as well. I don't know if they would have said that if I were white." [32]

> "When my mood is depressing, I sometimes think that if I can meet a few moms who are in the same trouble as me and talk to each other, I might feel better. If you can organize some of these events, I'd love to come!" [23]

*Sub-theme 2*: *Lack of practical support*. In the past, people tended to focus more on the emotional needs of mothers with postpartum depression, such as understanding, care, and encouragement, and easily overlooked the specific challenges and needs they faced in real life [15, 16,

27, 41]. Emotional support alone is often insufficient to solve the practical problems of post-partum depressed mothers caused by physiology, role change, and life stress. If family members can take the initiative to undertake more household chores, it will greatly reduce the burden on mothers and inhibit the development of postpartum depression.

> ". . .The first 40 days after you give birth, you go back to your mom's house. So, when I gave birth, I was at my mom's for 40 days. I had a lot of help from my mom and my sisters. They would cook the food." [15]

> "I wish he (my husband) could spend more time with me, chat with me, and take a walk with me, but he's too busy!" [27]

> "My husband is an engineer and is very research-based, so he really likes articles and what-not and likes to know the details of how the vaccination is created. He's really interested in that. I've really relied on him for that research piece of it." [16]

*Sub-theme 3*: *Lack of information support*. Information support can, to some extent, alleviate individual stress and reduce negative emotions. Some mothers wanted information and health education from professionals to help them with postnatal problems [15, 21–23, 27–30, 40], such as knowledge of postnatal depression, infant healthcare, breastfeeding skills, self-care, and body image recovery after childbirth.

> "I don't know anyone who has it (PND in Nigeria) because I don't have it. I didn't have it in Nigeria. I never knew it existed. . .do you understand? Until I came to this place." [30]

> "I wonder if the baby's jaundice is better. Is he still on the blue light? How long will the treatment take?" [28]

> "I had a cesarean section. I don't know if it's because of the anesthesia on my waist. After I came home from the hospital, I suddenly had to lie in bed for several days and couldn't move. My husband had to help me turn over, which was too painful. My mood was particularly low during those days. If I had known that I might be in such a situation, I would have paid attention to it myself!" [23]

> "I've heard that doing postnatal exercises can shape your body without harming it, but I don't know how to do it, so I really hope someone will guide me!" [27]

*Sub-theme 4*: *Lack of other support*. In addition to the lack of emotional support, practical support, and information support, some mothers may also lack support in terms of practical actions [15, 19, 25–30, 32, 34, 36, 40], such as financial support and job support. Some also report that they live in areas where there are no specialized mental health services to address psychological problems in a timely manner, which may cause misunderstanding among others.

> "suffer hardships" and "live hand to mouth." [29]

> "I was in charge of the company's human resources before, but now I can't go back to work for more than half a year, so I'm sure this piece will be handed over to him." [27]

> "There is a huge stigma of being mentally ill in the public, but for us Asians, there is a double disadvantage. I really fear that work will find out." [32]

## 4 Discussion

### 4.1 Patients with postpartum depression have prominent negative psychological experiences and bear heavier somatic burdens

Pregnancy is one of the most important life events in a woman's life. During pregnancy, childbirth, breastfeeding, and child rearing, a series of physiological and psychological changes can occur, such as crying, sadness, anxiety, emotional instability, confusion, loss of confidence in the future, postpartum uterine contraction pains, nipple pains, dizziness, wound pains, and insomnia. These changes can result in a prominent negative psychological experience, a heavier physical burden, and exacerbation of the postpartum depression development process [2]. Postpartum depression is very harmful, as it can affect the physical recovery of the mother, deteriorate the intimate relationship among family members, affect the secretion of breast milk, and even cause anxiety for the spouse of the mother, which may lead to the breakup of the family [6, 43, 44]. Severe postpartum depression can lead to incidents such as suicide and infanticide [2]. In the current study, it was also found that most of the patients with postpartum depression not only experienced physical and psychological discomfort but also had suicidal thoughts. Another study showed that negative emotional experiences of mothers can contribute to the development of social anxiety during school age [45]. This suggests that professionals and family members should pay attention to the physical and mental health of patients with postpartum depression and take appropriate measures in time to intervene in the treatment. This can include giving positive energy support to patients through words of guidance, books, film, and television materials to increase their confidence and hope in life, and to improve the negative psychological experience of patients with postpartum depression [27]. Some studies have shown that non-invasive treatments can be applied to improve the negative emotions and physical burden of patients. For example, postpartum acupressure can effectively stimulate acupoints, not only promoting milk secretion and maintaining the smooth flow of the breast, but also regulating the psychological state of the patient [46]. Music therapy can divert the patient's attention and, when combined with psychological incentives, to a certain extent, meet the patient's psychological needs and improve the outcome of labor [47]. Aromatherapy is a method of consciously controlling respiratory rhythm, allowing the patient to inhale neurochemicals with sedative and tranquilizing activity to regulate the autonomic nerves and achieve the effect of relieving mental stress and raising the pain threshold [48]. In addition, reasonable psychological care interventions can also effectively alleviate adverse emotions and improve the quality of life of mothers and infants. Their clinical efficacy has been universally certified, such as cognitive-behavioral therapy, interpersonal relationship therapy, and positive thinking interventions [2]. Finally, professionals can also instruct postpartum depressed patients to use deep breathing and emotional relaxation to control their emotions and reduce the incidence of emotional loss of control. The above methods are not only simple to operate but also easy to be accepted by postpartum depressed patients and their families.

### 4.2 Discomfort and impact of role transition in patients with postpartum depression

From pregnancy to childbirth, mothers experience rapid changes in their roles, and their psychological and emotional state is always in a state of serious instability, which is very likely to result in postpartum mother's role deficits and maladaptation [49]. With the change of the original lifestyle and family relationship, it may intensify the occurrence of family conflicts, which is not conducive to the recovery of postpartum depressed patients. In this study, several studies have shown that mothers cannot immediately enter the role of motherhood, cannot

fully adapt to the arrival of a new life, and are concerned about the impact on the health of the child. Therefore, it is important to let mothers experience the joy of motherhood as early as possible during pregnancy to gradually adapt to the role change and reduce the occurrence of postpartum role conflict [27]. For example, before delivery, hospitals can organize classes for pregnant women to encourage family members to participate, explaining the process of physiological changes in pregnancy, the prenatal examination system, and other knowledge. They can also organize practical classes on infant care to guide infant feeding and daily care, so as to promote early adaptation to the role of the mother. After delivery, continuity of care can be provided to avoid the formation of excessive dependency needs in the postpartum period. In addition, this study also found that some families are influenced by social and cultural factors, such as traditional practices like "doing the month," "women should stay at home to bring up children," and "gender favoritism." These practices may aggravate the psychological burden of mothers. Therefore, when providing preventive health care and therapeutic interventions for patients with postpartum depression, professionals should not only strengthen the education of patients' families but also pay attention to the multicultural differences of families. They should instruct family members to provide culturally appropriate support for mothers in order to reduce the risk of postpartum depression due to cultural differences [23].

## 4.3 Lack of relevant support is detrimental to recovery in patients with postpartum depression

Lack of relevant support is an important influence on postpartum depression [50–52], and a study by Cho et al. [53] showed that mothers with moderate or low levels of relevant support were more likely to suffer from postpartum depression. Support can be divided into formal support provided by the government, institutions, and grassroots communities, such as financial support provided by the government, and informal support provided by groups such as relatives, neighbors, and friends, such as emotional support, practical support, and information support [54]. Good supportive functioning can have a positive effect on buffering or minimizing the impact of adverse events on an individual's health. For example, "partner support," "close friend support," and "coworker support" can mitigate the effects of stress on one's physical and mental health [55]. In this study, it was found that most patients with postpartum depression hoped that their spouses would understand their psychological feelings and take the initiative to undertake household chores and participate in family activities such as infant rearing. And the family, as a basic social network in the environment, can buffer the mental stress of patients by providing support and information feedback when family members suffer from the disease [56]. A well-functioning family not only improves the mental health of patients with postpartum depression, maintains family intimacy, and mitigates the adverse effects of depression-induced stimulation or changes in pregnancy outcomes, but also contributes to their postpartum psychoemotional stabilization and promotes the quality of feeding for discharged infants [57]. It has also been shown that patients with postpartum depression who have good family functioning receive more emotional, behavioral, and informational support from their families, and their level of coping difficulties is lower [58].

At the same time, emotional support provided by professionals, information support, and practical action support provided by society are also important in alleviating negative emotions. In addition, there are also studies that show that prenatal-related support plays a crucial role in preventing postpartum depression [55]. Therefore, it is recommended that professionals should strengthen the longitudinal study of relevant support between the prenatal and postnatal periods. For example, an internet support platform can be set up in the prenatal period so that pregnant women and mothers can access a large amount of information about their

babies, emotional support, and even financial support, etc. During the hospitalization period in the postnatal period, a companion ward can be set up to promote communication between companions. After discharge from the hospital, postnatal patients can be given postnatal physical and psychological support through the implementation of kangaroo care or continuity of care in order to promote their role adaptation and improve their sense of competence in child rearing, and so on. Encourage postpartum depressed patients to communicate and confide in family members, instruct family members to take the initiative to undertake family tasks, and care for and look after the physical needs and emotional comfort of postpartum depressed patients, especially their partners. Let them feel the warmth and support of the family in order to reduce postpartum depression and adverse role experience, alleviate their symptoms such as tension, insomnia, irritability, crying, etc., and promote the generation of positive emotions. Finally, it is also possible to improve the system of comprehensive and standardized community-based maternal intervention management, with community health service institutions setting up outpatient obstetric psychological counseling clinics; actively providing financial assistance to economically disadvantaged mothers; and setting up convenient postnatal service stations, including postnatal physiotherapy, live-in nurse, and custodial services.

## 5 Limitations

This study has several limitations. First, only qualitative studies published in English or Chinese-indexed journals were selected based on the inclusion criteria of the literature. Some gray literature has not been searched, which may lead to biased information. In addition, the definition and specifics of postpartum depression may not be exactly the same in different countries and regions due to cultural and policy differences in each region. Of the 28 papers included, 19 did not mention the influence of the researcher's own values and cultural background. Different researchers have their own understandings and interests, and the psychological experiences and needs of people with postpartum depression are largely influenced by the cultures of different regions, which may lead to differences in results.

## 6 Conclusion

This study adopts qualitative research systematic integration and analysis methods, which, to some extent, integrate studies from different countries and medical contexts and can more realistically reflect the real psychological experience of postpartum depression patients. The integrated results indicate the prominence of negative physical and psychological experiences, the discomfort of role transformation, and the lack of relevant support in patients with postpartum depression. In future studies, more attention should be paid to maternal mental health, and full mental health screening during pregnancy and childbirth should be implemented, and psychological counseling services should be provided to pregnant women. At the same time, a hospital-community-family continuum of mental health service management system should be established, and a multidisciplinary team consisting of maternal and child health workers, mental health providers, and social workers should be set up to carry out collaborative care models, so as to provide better healthcare services for women during pregnancy and childbirth.

## Supporting information

**S1 File. Search formula using PubMed as an example.**
(DOCX)

**S2 File. Data extraction and integration process.**
(DOCX)

## Author Contributions

**Data curation:** Wu Jiaming, Guo Xin, Du Jiajia, Peng Junjie.

**Investigation:** Guo Xin, Peng Junjie.

**Methodology:** Wu Jiaming, Guo Xin, Du Jiajia, Peng Junjie, Hu Xue, Li Yunchuan, Wu Yuanfang.

**Project administration:** Hu Xue.

**Resources:** Wu Jiaming, Hu Xue.

**Software:** Peng Junjie.

**Supervision:** Wu Jiaming, Guo Xin, Du Jiajia, Hu Xue.

**Validation:** Wu Jiaming, Guo Xin, Du Jiajia, Peng Junjie, Hu Xue, Li Yunchuan, Wu Yuanfang.

**Visualization:** Li Yunchuan, Wu Yuanfang.

**Writing – original draft:** Wu Jiaming.

**Writing – review & editing:** Wu Jiaming, Guo Xin.

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
