## [Decision Letter · Decision Letter 0]

18 Sep 2024

PONE-D-24-08898Psychological Experience of Patients with Postpartum Depression:A qualitative Meta-synthesisPLOS ONE

Dear Dr. xue,

Thank you for submitting your manuscript to PLOS ONE. After careful consideration, we feel that it has merit but does not fully meet PLOS ONE’s publication criteria as it currently stands. Therefore, we invite you to submit a revised version of the manuscript that addresses the points raised during the review process.

The manuscript must be returned for major corrections. For the following reasons: The prism flowchart used has four stages, the one currently used has 3 stages, consult Prisma (https://www.prisma-statement.org/prisma-2020-flow-diagram) since it is not a frequent analysis method, it is important that the authors detail the data extraction process to arrive at three main themes and how subtopics arise within them. So that anyone can reproduce the study. In turn, since a psychometric measure is used for the discrimination of women with depression, the cultural factors evidenced in the limitations of the study require greater precision, since the psychometric instrument has validity and reliability independent of the cultural group. In the discussion, since it is a study that favors the deep understanding of depression, it is important to discuss each of the topics found. These topics should be discussed about the implications they have for the current scientific community, public policy-making, and women and family members so that they know how they can use the results of the study. In the table of results it is important to homogenize the presentation of the last study, it does not have a synthesis of results as they have been presenting in the previous studies in the table

We look forward to receiving your revised manuscript.

Kind regards,

Alejandro Botero Carvajal, MD

Academic Editor

PLOS ONE

“This study is supported by the Science and Technology Innovation Fund from the School of Nursing, Yunnan University of Chinese Medicine(YZHCKY2406)”

5. Please remove your figures from within your manuscript file, leaving only the individual TIFF/EPS image files, uploaded separately. These will be automatically included in the reviewers’ PDF.

6. Please include your tables as part of your main manuscript and remove the individual files. Please note that supplementary tables (should remain/ be uploaded) as separate "supporting information" files

Please see the comments below:

Reviewers' comments:

Reviewer's Responses to Questions

**Comments to the Author**

1. Is the manuscript technically sound, and do the data support the conclusions?

Reviewer #1: Yes

Reviewer #2: Yes

2. Has the statistical analysis been performed appropriately and rigorously? 

Reviewer #1: Yes

Reviewer #2: Yes

3. Have the authors made all data underlying the findings in their manuscript fully available?

Reviewer #1: Yes

Reviewer #2: No

4. Is the manuscript presented in an intelligible fashion and written in standard English?

Reviewer #1: Yes

Reviewer #2: Yes

5. Review Comments to the Author

Reviewer #1: Introduction:

The introduction provides a comprehensive view of postpartum depression, which is a very common health problem among postpartum women worldwide, causing distress and serious negative impacts on mothers and their families, and has received extensive attention in several countries. This effectively contextualizes the importance of conducting a recent systematic review on the psychological experiences of patients with postpartum depression using qualitative studies.

Methods:

Its methods section is good for its clarity and systematic approach. The procedures for the systematic search (protocol and registry, systematic search, study selection, data extraction and quality assessment, data synthesis and analysis) are meticulously detailed.

Results:

The presentation of results is clear and concise, offering a step-by-step account of the articles selected for systematic review (selection and characteristics of trials, primary results, secondary results).

Discussion:

Their discussion effectively connects the findings of their study to previous research on qualitative studies on the psychological experience of postpartum depressed patients. Her discussion addresses topics on pregnancy, the physiological and psychological factors that produce changes in the maternal body, of which postpartum depression is one of the most common health problems among postpartum women around the world.

Limitations of the study.

The authors mention as a limitation qualitative studies published only in journals indexed in English or Chinese according to the inclusion criteria and another limitation is the specific details of postpartum depression, which may be different between countries.

Conclusion.

In summary, the most relevant finding revealed that the impact of negative physical and psychological aspects, the discomfort of role transformation and the notable lack of support in patients with postpartum depression, specifies the lack of more studies to address mental health during pregnancy.

The article is accepted for publication.

Reviewer #2: Thank you for the opportunity to review this manuscript: “Psychological Experience of Patients with Postpartum Depression: A qualitative Meta-synthesis”

This manuscript provides an important synthesis of the literature around postnatal depression and the key themes that warrant further consideration in formulating strategies for supporting women suffering PND. I strongly support the publication of this article, however, while the study itself was conducted well, I feel that the extracted findings were inadequately discussed and synthesized. Importantly, I’m concerned that there are important subthemes that may be missing. Also, given the inclusion of both Chinese and International studies, I am interested in knowing whether there are sociocultural determinants of PND in Asian and Western contexts, and whether/how these might influence the recommendations the authors might make to improve PND outcomes in these very different contexts. In light of these issues I am recommending this manuscript as ACCEPT WITH MAJOR REVISIONS to allow the authors time to address these concerns and to make the necessary changes. I am looking forward to seeing the revised manuscript in due course

Thanks and best wishes.

OVERALL

Language is mostly good throughout but a few spelling and grammatical errors throughout. Some of the spelling errors are in quotes so not sure what to do about those.

Do you have supplementary material for this study? It would be useful to be able to inspect the quotes extracted from each paper and detail on how they were grouped into themes.

ABSTRACT

May need to be updated following changes to the results/discussion.

INTRODUCTION

Remove “etc.” throughout. Scientific articles should be as specific as possible.

Some studies have reported that the global average incidence of postpartum depression is 17.7%[4] – I read this paper and could not find this statistic. Could you please clarify?

METHODS

One issue here is that the included literature is now 9 months out of date. Would be great if the authors could update the search to September 2024. I can’t imagine this would identify too many new articles but it would give the most comprehensive snapshot of the literature.

Section 2.3 – the example search term for PubMed isn’t necessary and you could move this to supplemental.

RESULTS

Can you briefly describe what "doing the month" is? I see it is mentioned in the quote further down but a brief description above would be valuable for non-Chinese readers.

I’m not familiar with this specific form of analysis but I am wondering whether it would be appropriate to calculate and describe which were the themes or subthemes that appeared most frequently across many studies? This might give an indication which themes are “universal” across different social and cultural contexts, or even common within cultures (e.g., Asian vs Western), which may give an indication as to which themes or subthemes should be addressed as a matter of priority.

I am wondering if it is appropriate to broaden the range of themes and subthemes identified. I can imagine emotional experiences, cognitive experiences, physical experiences, social and interpersonal experiences, behavioural changes (in self and others), identity and role transition, support experiences (friends, family, healthcare), coping strategies, feelings about impact of parenting on baby, cultural and societal influences amongst others as being key contributors. I note that you’ve included many of these but I am wondering if there are more themes that might have significant evidence. One of the key domains that I think could be expanded are around meta-theme 3 – support. For example, there seemed to be a several comments that could suggest that another relevant sub-theme could be “lack of practical support” (as distinct from “other support”) – I could see this being a strong contributor to PND having a husband or mother-in-law that won’t provide practical support for domestic duties. I could also see stronger recommendations around the expectations that husbands and other family members support new mothers post partum as being particularly helpful in alleviating some of the main modifiable physical stressors.

DISCUSSION

The first paragraph justifying the need for qualitative approach is redundant. If necessary you could include some of this detail in the justification in the introduction? Otherwise I would just start the discussion by summarising your main findings.

I think the discussion could benefit from some more structure and depth. You’ve just synthesized some very interesting literature but the discussion as written missed a significant opportunity to make sense of these findings and propose some steps forward. As a starting point, I think that subheadings for each of the 3 main meta-themes would be helpful. I think under these headings it would be useful to 1) summarise the findings from this meta-synthesis, and 2) provide recommendations for how these might inform solutions and how these might reduce the risk of PND.

I agree with you completely regarding the role of support. I believe this is a critical component and I would like to see this expanded in the discussion. People generally fare much better in life when they feel supported and informed, both practically and emotionally, and this is particularly the case with new mothers. I would like to see more in depth discussion and recommendation around this point.

As mentioned above, it would be useful to know how person-centred care might be applied across cultural contexts – Asian vs Western contexts.

6. PLOS authors have the option to publish the peer review history of their article (what does this mean?). If published, this will include your full peer review and any attached files.

Reviewer #1: **Yes: **Horacio Almanza-Reyes

Reviewer #2: No

---

## [Author Response · Author response to Decision Letter 0]

4 Oct 2024

Dear Editor and Reviewers:

 Greetings!

 First of all, on behalf of all the authors, please allow me to extend our sincerest thanks to you and the reviewers. We would like to thank you for taking the time out of your busy schedules to review our paper in detail and provide valuable comments and suggestions. We are fully aware of the importance of these comments in improving the quality of the paper and ensuring academic rigor.

After receiving the review comments, we immediately organized our team to conduct in-depth discussion and analysis, and revised and improved the paper in strict accordance with the reviewers' suggestions. The following are our responses and explanations to the reviewers' comments:

Editorial comment：

1.The prism ﬂowchart used has four stages, the one currently used has 3 stages, consult Prisma(https://www.prisma-statement.org/prisma-2020-ﬂowdiagram)：

 Thank you to the editors for their valuable comments on "The prism flowchart," which previously used a three-stage flowchart but has now been changed to four stages, incorporating reviewer 2's comments. （One issue here is that the included literature is now 9 months out of date. It would be great if the authors could update the search to September 2024.） The literature search was updated, and the corresponding results of the flowchart were modified, as shown in Figure 1.(Fig. 1)

2.Since it is not a frequent analysis method, it is important that the authors detail the data extraction process to arrive at three main themes and how subtopics arise within them. So that anyone can reproduce the study:

 We strongly agree with the editor's comments because meta-synthesis of qualitative studies is an uncommon method of analysis. Therefore, at the editor's suggestion, we have provided a detailed description of the data extraction process in this study, including how the three major themes were derived and how sub-themes were generated within these themes. The description is as follows:

 We used meta-aggregation to synthesize the results of the qualitative study. The results of the literature were integrated using the pooled integration method recommended by the JBI Center for Evidence-Based Health Care. Guided by qualitative research methods, researchers repeatedly read, deeply dissected, and interpreted the findings of the included literature and formed new sub-themes after combining similar findings. The sub-themes with certain connections were then synthesized into a new integrative theme, and the corresponding sub-themes were assigned to the integrative theme. The two researchers repeatedly read, analyzed, and compared 28 pieces of literature to distill a total of 51 findings. They grouped similar findings into 12 sub-themes and synthesized them into three descriptive themes, with no mutual exclusivity in the categorization of findings(S2 File)

3.In turn, since a psychometric measure is used for the discrimination of women with depression, the cultural factors evidenced in the limitations of the study require greater precision, since the psychometric instrument has validity and reliability independent of the cultural group:

 We strongly agree that the validity and reliability of psychometric instruments are independent of cultural groups. However, the differences in cultural factors are also described more specifically in the study limitations. For example, "Of the 28 papers included, 19 did not mention the influence of the researcher's own values and cultural background."

4.In the discussion, since it is a study that favors the deep understanding of depression, it is important to discuss each of the topics found. These topics should be discussed about the implications they have for the current scientiﬁc community, public policy-making, and women and family members so that they know how they can use the results of the study:

 We are very grateful to the editors for raising deficiencies regarding the discussion section. This study comprehensively collects qualitative studies related to the psychological experience of patients with postpartum depression. It further analyzes and summarizes the results of these studies and comprehensively interprets the psychological feelings and experiences of patients with postpartum depression. This will provide a reference basis for clinical workers to develop relevant nursing strategies. However, the previous discussion of logic and analysis is insufficient. At the suggestion of the editor and reviewers, we have made significant modifications to the discussion section and categorized the discussion around the findings of the study into three major themes with detailed comments (4.1 Patients with postpartum depression have prominent negative psychological experiences and bear heavier somatic burdens; 4.2 Discomfort and impact of role transition in patients with postpartum depression; 4.3 Lack of relevant support is detrimental to recovery in patients with postpartum depression), see manuscript for details 4. Discussion section. We believe that the impact of these themes on the current scientific community, public policy development, women, and family members will allow other scholars to capitalize on the research findings.

5.In the table of results it is important to homogenize the presentation of the last study, it does not have a synthesis of results as they have been presenting in the previous studies in the table:

 We are very grateful to the editors for identifying this problem, and as soon as we found it, we synthesized the last result in the results table, which has been modified to read "Loneliness; Hope; Obsessive thoughts ; Haunted ; Loss of interest and purpose. Fear and guilt; Unable to concentrate; Negativity; Insanity; Loss of control of emotions; Insecurities" eleven themes.

6.If you would like to make changes to your ﬁnancial disclosure, please include your updated statement in your cover letter. Guidelines for resubmitting your ﬁgure ﬁles are available below the reviewer comments at the end of this letter：

 Thanks to the editor's proposal, we have decided, after consideration, not to make changes to the financial disclosure.

7.If applicable, we recommend that you deposit your laboratory protocols in protocols.io to enhance the reproducibility of your results:

 Thanks to the editors for the suggestion, but after careful consideration we decided not to save the experimental procedure on Protocols.io.

8.Please ensure that your manuscript meets PLOS ONE's style requirements, including those for ﬁle naming:

 In order to comply with the strict formatting requirements of PLOS ONE, we have revised the formatting of the previous manuscript, which now conforms to PLOS ONE style, as described under "Revised Manuscript with Track Changes" and "Manuscript."

9.Please state what role the funders took in the study. If the funders had no role, please state: "The funders had no role in study design, data collection and analysis, decision to publish, or preparation of the manuscript." If this statement is not correct you must amend it as needed. Please include this amended Role of Funder statement in your cover letter; we will change the online submission form on your behalf：

 We have taken a look at the role of the funder in the research and have attached a revised funder role statement to the cover letter. The statement is: "This study is supported by the Science and Technology Innovation Fund from the School of Nursing, Yunnan University of Chinese Medicine (YZHCKY2406). Yunnan University of Traditional Chinese Medicine School of Nursing provided financial support for this study."

10.We note that your Data Availability Statement is currently as follows: [All relevant data are within the manuscript and its Supporting Information ﬁles.] Please conﬁrm at this time whether or not your submission contains all raw data required to replicate the results of your study. Authors must share the “minimal data set” for their submission. PLOS deﬁnes the minimal data set to consist of the data required to replicate all study ﬁndings reported in the article, as well as related metadata and methods：

 In accordance with the requirements of PLOS ONE's Data Availability Statement and the recommendations of the editors, we have examined all of the raw data used in this study and assure you that our paper contains all of the raw data necessary to replicate the results of the study, as described in Table 1 of the manuscript and in the Supporting Information (S2 File).

11.PLOS requires an ORCID iD for the corresponding author in Editorial Manager on papers submitted after December 6th, 2016. Please ensure that you have an ORCID iD and that it is validated in Editorial Manager:

 In accordance with the PLOS requirement that corresponding authors of papers submitted after December 6, 2016, provide an ORCID iD in Editorial Manager, we double-checked the ORCID iD of the corresponding author of this paper, Xue Hu, and ensured that she already had an ORCID iD (0009-0003-4439-5737) that had been verified in Editorial Manager.

12.Please remove your ﬁgures from within your manuscript ﬁle, leaving only the individual TIFF/EPS image ﬁles, uploaded separately. These will be automatically included in the reviewers’ PDF:

 We have strictly followed the editor's advice to remove the images from the manuscript and upload the images separately, as described under "Revised Manuscript with Track Changes" and " Manuscript."

13.Please include your tables as part of your main manuscript and remove the individual ﬁles. Please note that supplementary tables (should remain/ be uploaded) as separate "supporting information" ﬁles:

 We have strictly followed the editor's advice to include the forms as part of the main manuscript and remove the individual files. A new table (Table 2) has also been added, taking into account the comments of reviewer 2, which has been included in the manuscript and uploaded as a separate "Supporting Information" file.

Reviewer 1:

 Regarding Reviewer 1's comments, we are very grateful to the reviewer for taking the time to read our paper, and it is an honor to receive Reviewer 1's recognition and praise of our paper. In addition, our careful reading of Reviewer 1's comments does not appear to suggest any changes, so we have not revised Reviewer 1's comments.

Reviewer 2:

 Regarding Reviewer 2's comments, we would like to thank the reviewer for taking the time to read our paper and for giving us the opportunity to revise it with many meaningful comments. We have carefully read Reviewer 2's comments and revised each of them in order to take this opportunity to improve the rigor and scientific quality of our paper, as shown in the following responses:

1.Language is mostly good throughout but a few spelling and grammatical errors throughout. Some of the spelling errors are in quotes so not sure what to do about those：

 We couldn't agree with you more. We did find some grammatical and spelling mistakes after we read the paper carefully, so we asked a professional English expert to guide us on the grammar and spelling of our paper. After the guidance, we carefully revised the entire manuscript with appropriate adjustments. Please refer to the "Manuscript with Track Changes" and "Manuscript." 

2.Do you have supplementary material for this study? It would be useful to be able to inspect the quotes extracted from each paper and detail on how they were grouped into themes：

 Because meta-synthesis of qualitative studies is an uncommon method of analysis, we have provided a detailed description of the data extraction process in this study, including how the three major themes were derived and how sub-themes were generated within these themes, at the editor's suggestion. The description is as follows:

 We used meta-aggregation to synthesize the results of the qualitative study. The results of the literature were integrated using the pooled integration method recommended by the JBI Center for Evidence-Based Health Care. Guided by qualitative research methods, researchers repeatedly read, deeply dissected, and interpreted the findings of the included literature and formed new sub-themes after combining similar findings. The sub-themes with certain connections were then synthesized into a new integrative theme, and the corresponding sub-themes were assigned to the integrative theme. The two researchers repeatedly read, analyzed, and compared 28 pieces of literature to distill a total of 51 findings. They grouped similar findings into 12 sub-themes and synthesized them into three descriptive themes, with no mutual exclusivity in the categorization of findings. See the Supporting Information (S2 File), where we strongly agree with the reviewers that the Supporting Information will be more helpful to others in understanding postpartum depression.

3.ABSTRACT：May need to be updated following changes to the results/discussion：

 Thanks to the reviewer's reminder, we made appropriate changes to the methods and results of the abstract after revising the content of the paper, as described in "Manuscript with Track Changes" and "Manuscript ."

4.INTRODUCTION：Remove “etc.” throughout. Scientiﬁc articles should be as speciﬁc as possible：

 We have removed "etc." from the preface in strict accordance with the reviewer's comments to ensure specificity of the scientific article.

5.INTRODUCTION：Some studies have reported that the global average incidence of postpartum depression is 17.7%[4] – I read this paper and could not ﬁnd this statistic. Could you please clarify?：

 First of all, we are very grateful to the reviewers for their scientific rigor, and we are very sorry for this, as our lack of rigor and conscientiousness has caused misunderstanding to the reviewers. After we found this problem, we immediately verified the relevant data and indeed did not find this data in this article. After we realized the seriousness of the matter, we immediately reorganized the literature referenced in this paper and found the cause of the error (due to our lack of seriousness, the references were cited incorrectly). After discovering this error, we immediately revised the relevant data and literature, which is now revised to "17.2%", and updated the references (4.Wang Z, Liu J, Shuai H, Cai Z, Fu X, Liu Y, et al. Mapping global prevalence of depression among postpartum women[J]. Transl Psychiatry. 2021 Oct 20; 11(1):543. https://doi: 10.1038/s41398-021-01663-6). We apologize again for this.

6.METHODS：One issue here is that the included literature is now 9 months out of date. Would be great if the authors could update the search to September 2024. I can’t imagine this would identify too many new articles but it would give the most comprehensive snapshot of the literature：

 We strongly agree with the reviewers that the long submission cycle has resulted in the inclusion of literature that is now 9 months out of date. In order to ensure that the results of the meta-synthesis are scientifically sound, comprehensive, and up-to-date, we have strictly followed the reviewers' comments and conducted a new search of the nine previous databases from January 1, 2024, to September 20, 2024, using the same methodology used for the previous literature search of this dissertation. Interestingly, 154 new articles were found (PubMed (n=18); Web of Science (n=98); Embase (n=16); The Cochrane Library (n=21); Medline (n=1); CNKI (n=0); WANGFANG(n=0); VIP(n=0); SinoMed(n=0)). Unfortunately, on new articles were found in the four Chinese databases. After we screened and read the 154 articles strictly according to the inclusion and exclusion criteria of this dissertation, we finally included 2 papers, the first and second papers in Table 1 (“Hanach et al [15], 2024”; “Borrero et al [16], 2024"). And 2 new sub-themes were identified from these two literatures ("Impact of parenting"; "Lack of practical support"). See specifically "Manuscript with Track Changes" and "Manuscript ."

7.METHODS：Section 2.3 – the example search term for PubMed isn’t necessary and you could move this to supplemental：

 Thanks to the reviewers' comments, we have removed the example search terms in Section 2.3 - PubMed from the manuscript and moved them to the Supplementary 

---

## [Editor Report · Decision Letter 1]

17 Oct 2024

Psychological Experience of Patients with Postpartum Depression:A qualitative Meta-synthesis

PONE-D-24-08898R1

Dear Dr. xue,

We’re pleased to inform you that your manuscript has been judged scientifically suitable for publication and will be formally accepted for publication once it meets all outstanding technical requirements.

Kind regards,

Alejandro Botero Carvajal, MD

Academic Editor

PLOS ONE
---

## [Editor Report · Acceptance letter]

28 Oct 2024

PONE-D-24-08898R1 

PLOS ONE

Dear Dr. Xue, 

I'm pleased to inform you that your manuscript has been deemed suitable for publication in PLOS ONE. Congratulations! Your manuscript is now being handed over to our production team.

Kind regards, 

on behalf of

Dr. Alejandro Botero Carvajal 

Academic Editor

PLOS ONE